# WaveAR: Wavelet-Aware Continuous Autoregressive Diffusion for Accurate Human Motion Prediction

**Shengchuan Gao**[1*]    **Shuo Wang**[2*]    **Yabiao Wang**[2,3]    **Ran Yi**[1†]

[1]Shanghai Jiao Tong University    [2]Tencent Youtu Lab    [3]Zhejiang University

`gscdy111@sjtu.edu.cn, leifwang@tencent.com`
`caseywang@tencent.com, ranyi@sjtu.edu.cn`

## Abstract

This work tackles a challenging problem: stochastic human motion prediction (SHMP), which aims to forecast diverse and physically plausible future pose sequences based on a short history of observed motion. While autoregressive sequence models have excelled in related generation tasks, their reliance on vector-quantized tokenization limits motion fidelity and training stability. To overcome these drawbacks, we introduce **WaveAR**, a novel AR based framework which is the first successful application of a continuous autoregressive generation paradigm to HMP to our best knowledge. WaveAR consists of two stages. In the first stage, a lightweight Spatio-Temporal VAE (ST-VAE) compresses the raw 3D-joint sequence into a downsampled latent token stream, providing a compact yet expressive foundation. In the second stage, we apply masked autoregressive prediction directly in this continuous latent space, conditioning on both unmasked latents and multi-scale spectral cues extracted via a 2D discrete wavelet transform. A fusion module consisting of alternating cross-attention and self-attention layers adaptively fuses temporal context with low- and high-frequency wavelet subbands, and a small MLP-based diffusion head predicts per-token noise residuals under a denoising loss. By avoiding vector quantization and integrating localized frequency information, WaveAR preserves fine-grained motion details while maintaining fast inference speed. Extensive experiments on standard benchmarks demonstrate that our approach delivers more accurate and computationally efficient predictions than prior state-of-the-art methods.

## 1   Introduction

Human Motion Prediction (HMP) involves forecasting future human poses or motions from an observed sequence of historical poses. This capability not only deepens our understanding of human behavior patterns but also underpins a wide range of applications [10, 22, 31, 41, 51, 54, 55, 57, 60, 62]—autonomous driving [23, 39], robotics [16], human–computer interaction [25], virtual reality [13, 24, 30], and assisted healthcare [44]—making HMP a rapidly advancing frontier in computer vision and artificial intelligence interaction. Early HMP methods were largely deterministic[8, 28, 34], which predicted the single most likely future trajectory. While effective in some settings, these approaches neglect the inherent uncertainty and fail to capture the rich diversity of possible action sequences. Recently, the field has increasingly embraced stochastic generative frameworks such as variational autoencoders (VAEs)[9, 35, 58], generative adversarial networks (GANs)[19], and diffusion-based models[6, 43]. In this work, we focus on the Stochastic Human Motion Prediction (SHMP) problem that targets generating diverse yet accurate future pose sequences conditioned on the observed history.

---

[*]Equal contribution
[†]Corresponding author

39th Conference on Neural Information Processing Systems (NeurIPS 2025).

Traditional variational autoencoder (VAE) and generative adversarial network (GAN) methods, while effective at modeling uncertainty and diversity, can sometimes produce futures that contradict the observed history—even generate abrupt, physically implausible transitions[61]. Diffusion-based approaches yield high-fidelity trajectories but incur substantial computational overhead that hampers real-time applicability. Autoregressive (AR) modeling, in contrast, generates each future pose sequentially conditioned on past context, inherently preserving temporal coherence, capturing multimodal uncertainty, and producing faithful, diverse outputs. However, existing AR formulations have two key limitations: (1)Most AR models rely on vector quantization (VQ) to discretize continuous motion sequences into finite token sets, which can introduce quantization artifacts and training instability (e.g., codebook collapse) that degrade motion fidelity and continuity, as analyzed in recent studies[18][61] ; (2) They lack the capacity to represent fine-grained dynamics such as sudden accelerations and intricate motion transitions. These issues motivate us to design a continuous, quantization-free AR paradigm for motion modeling. Besides, prior works have also attempted to leverage frequency-domain representations—most commonly via the Discrete Cosine Transform (DCT)—to capture temporal patterns, but DCT only preserves low-frequency content and omits high-frequency content that captures motion details, thus impairing prediction accuracy.

Recent studies have shown that images can be generated with pure autoregressive models—omitting vector quantization entirely[29]—and in doing so suggest novel directions for AR-based generative methods[11, 49]. In this paper, we propose **WaveAR** (Fig 1), a novel AR based framework for human motion prediction that operates entirely in continuous space. This continuous formulation naturally aligns with the temporal characteristics of motion sequences, enabling the model to capture long-range dependencies more effectively and maintain stable training dynamics. Specifically, it consists of two stages: In the first stage, we employ a Spatial-Temporal Variational AutoEncoder which simultaneously captures both spatial and temporal dependencies. This VAE can project the observed motion sequence into a smooth latent embedding in a continuous space, eliminating the quantization errors inherent in token-based schemes. In the second stage, we perform step-wise autoregressive forecasting over these latents, while simultaneously extracting low- and high-frequency components via multiscale discrete wavelet transforms (DWT) to guide each prediction. A fusion module consisting of alternating cross-attention and self-attention layers adaptively merges the time-domain latents with their wavelet-derived counterparts, preserving the VAE's learned trajectory trends and injecting sharp, transient motion details—yielding future pose sequences that are both temporally coherent and richly expressive.

Our contributions can be summarized as follows:

- We propose WaveAR, a novel framework for stochastic human motion prediction based on a continuous autoregressive paradigm that avoids quantization artifacts and effectively captures long-range temporal dependencies for accurate forecasting.
- We design a Masked Autoregressive Diffusion module with Wavelet Guidance, where multi-scale wavelet subbands are extracted by DWT and fused with masked future latents via alternating cross-attention and self-attention layers. Afterwards, a compact MLP-based diffusion predicts per-token noise distribution under a denoising loss.
- We introduce a lightweight Spatio-Temporal VAE (ST-VAE) that temporally downsamples the raw 3D-joint sequence into latent tokens, preserving joint structure while reducing sequence length for efficient downstream modeling.
- We validate WaveAR on standard HMP benchmark datasets: Human3.6M and HumanEva-I. Both quantitative and visualization results show that our method achieves more accurate performance with faster inference time compared to the state-of-the-art baselines.

## 2 Related Works

**Human Motion Prediction.**    Early human motion prediction methods were predominantly deterministic, producing a single "most likely" future trajectory. Most of these frameworks cast forecasting as a direct mapping from past poses to a fixed-length future sequence, using architectures such as RNNs [21, 36] or self-attention Transformers [1, 37, 46]—often enhanced with graph convolutional layers [8, 27] to capture spatial dependencies among joints. However, all of these solutions disregard the intrinsic uncertainty of human behavior. The advent of deep generative models has catalyzed a paradigm shift toward stochastic prediction frameworks through three principal avenues:

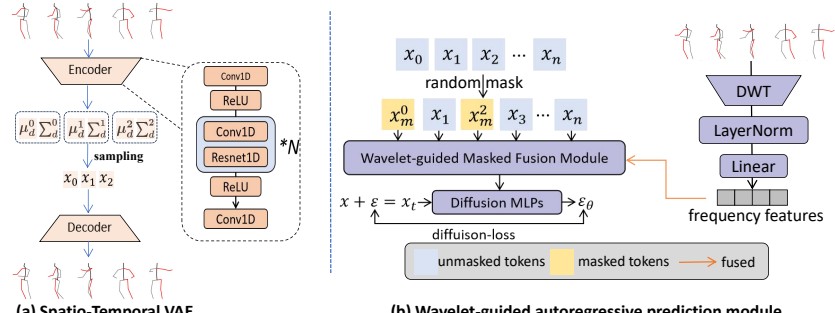

Figure 1: **Overall architecture of our proposed WaveAR.** (**a**) During training, a lightweight Spatio-Temporal VAE encodes the raw 3D-joint sequence (past $H$ + future $F$ frames) into a compact latent token stream via temporal downsampling. (**b**) shows the process of the wavelet-guided autoregressive masked generation model. First, the VAE latents are randomly masked, while the original input sequence's history undergoes a 2D discrete wavelet transform for wavelet frequency-domain feature extraction, and linear projection into the same embedding space. Next, the masked latents and projected wavelet features are fused through a fusion module consisting of alternating cross-attention and self-attention layers. Finally, a compact MLP-based diffusion predictor takes the autoregressive model's output as a conditioning vector and estimates the noise residual for each token, modeling its diffusion distribution and acting as a prediction head.

(1) **GAN-based** methods [3, 19, 26, 56] which generate diverse trajectories through adversarial training, (2) **VAE-based** methods [5, 9, 14, 35, 47, 53, 57, 58] that encode motion multimodality via latent distributions and most recently (3) **Diffusion based** methods [2, 6, 7, 43, 48, 50]. Among these methods, CoMusion [43] integrates a single-stage diffusion model with Transformer reconstruction and GCN refinement in DCT space for history-consistent stochastic forecasting; SkeletonDiffusion [7] introduces nonisotropic Gaussian diffusion via typed-graph convolutions with skeleton-aware noise covariance; BeLFusion [2] employs conditional latent diffusion [40] by sampling disentangled behavior codes to drive motion; HumanMAC [6] reframes prediction as masked DCT-based diffusion completion, jointly denoising observed and future frames for controllable, diverse outputs. Neuroscientific studies have shown that human motion exhibits strong frequency-domain characteristics, which benefit tasks such as motion editing and motion synthesis. Accordingly, many of the aforementioned motion-prediction approaches exploit frequency-domain representations—particularly the Discrete Cosine Transform (DCT)—to model motion distributions. MotionWavelet [12] applies wavelet manifold learning to motion prediction. Inspired by these findings, we integrate wavelet-domain cues into our latent-space prediction pipeline, resulting in more accurate and temporally coherent future poses.

**Autoregressive Modeling without Vector Quantization.** Traditional autoregressive models for sequential data generation, such as images or motion, heavily rely on discrete tokenization via vector quantization (VQ) [45], which introduces quantization artifacts and training instability. Different from vector-quantization based AR models, which represent the probability distribution of each token as a discrete multinomial distribution through a VQ codebook, MAR [29] circumvents these discrete representations' limitations by operating directly in continuous-valued spaces and representing the probability distribution of each token through a diffusion process. Such continuity avoids the loss of information during quantization, thus greatly enhances its generative quality. Building on MAR's continuous, quantization-free foundation, a growing body of work has applied its framework across diverse domains. MARRS [49] extends MAR's autoregressive framework to action–reaction synthesis by replacing discrete codecs with continuous representations and variational sampling. DART [15] extends MAR's diffusion-driven autoregressive framework to a non-Markovian paradigm, enabling high-resolution image synthesis in a unified Transformer sequence. MARDM [38] extends MAR's masked autoregressive diffusion framework to text-conditioned motion synthesis, restructuring motion latents via a bidirectional masked diffusion.

# 3 Method

## 3.1 Problem Formulation

Given an observed sequence of human motion with $P$ historical poses, denoted as $\mathbf{X} = [\mathbf{p}_{-P+1}, \mathbf{p}_{-P+2}, \ldots, \mathbf{p}_0] \in \mathbb{R}^{P \times J \times 3}$, where $\mathbf{p}_t$ represents the 3D coordinates of $J$ body joints at timestep $t$, the goal of Human Motion Prediction (HMP) is to forecast the subsequent $F$ future poses $\mathbf{Y} = [\mathbf{p}_1, \mathbf{p}_2, \ldots, \mathbf{p}_F] \in \mathbb{R}^{P \times J \times 3}$. For **Stochastic Human Motion Prediction (SHMP)**, the objective extends to generating $N$ plausible future trajectories $\widetilde{Y} = \{Y_1, Y_2, \ldots, Y_N\} \in \mathbb{R}^{N \times F \times J \times 3}$, where each $Y_i$ maintains temporal coherence while exhibiting distinct motion patterns. The key challenges in HMP include improving prediction accuracy for long-term motions, avoiding unnatural movements like sudden joint twists or behaviours that break physical laws.

## 3.2 Overview of WaveAR

Our method is the first framework for stochastic human motion prediction that employs an autoregressive model within a continuous space encoding paradigm. It consists of two stages to predict future poses from $H$ observed frames: First, we encode the input sequence into a smooth latent embedding using a lightweight **Spatio-Temporal VAE (ST-VAE)**. This step reduces the input dimension and creates continuous-valued tokens. In parallel, the raw 3D-joint sequence is fed into our Wavelet Feature Extractor, which applies a 2D discrete wavelet transform to generate four spectral subbands. These subbands help extract finer details in the motion dynamics, which are then linearly projected into the same embedding space as the latents, enabling more accurate predictions. We then recover the $F$ future tokens with our **Masked Autoregressive Diffuser**: at each iteration, a subset of tokens is masked, and the **Wavelet guided fusion module** fuses spectral cues (keys/values) with token queries to get latent motion information. It is used to denoise the masked positions through several MLP layers together with a diffusion loss. After all tokens are restored, a Latent Decoder reconstructs the full 3D joint trajectories. The overall structure of our network is shown in Fig. 1.

## 3.3 Spatio-Temporal VAE

To eliminate the dependency of prior autoregressive approaches on discrete VQ-VAE codes, we employ a lightweight Spatio-Temporal VAE (ST-VAE) that produces fully continuous latent tokens, simultaneously obtaining a vector representation that integrates both temporal and spatial information . In the first stage, we use this ST-VAE (see Fig. 1(a)) to compress the raw 3D-joint trajectory—comprising $H$ past frames and $F$ future frames—into a shorter sequence of continuous latents. Specifically, given a mini-batch $X \in \mathbb{R}^{B \times T \times C}$, where $T$ is the sum of the historical and predicted frames, and $C$ is the number of joints multiplied by 3 (representing the 3D position vector at each time step), we reshape it to $B \times C \times T$ and pass it through the encoder, which consists of a 1D convolution, several ResNet1D blocks [59], whose role is to integrate spatial information, and strided convolutions that downsample time by a factor $r$ to effectively integrate temporal information. The encoder then outputs $\mu$ and $\sigma$ for each latent token, and we sample via the reparameterization trick:

$$\mathbf{z} = \mu(\mathbf{X}) + \sigma(\mathbf{X}) \odot \epsilon, \quad \epsilon \sim \mathcal{N}(0, I), \tag{1}$$

where $\mu^{1:L}$ and $\epsilon^{1:L}$ are output of the encoder. To get the reconstructed original sequence $\hat{X}$, $Z$ is first passed through a post-quantization and then through the Decoder. The ST-VAE is trained to minimize the following loss:

$$\mathcal{L}_{\text{VAE}} = \|\hat{\mathbf{X}} - \mathbf{X}\|_1 + \beta \, \text{KL}\big(\mathbf{q}(\mathbf{Z} \mid \mathbf{X}) \,\|\, \mathcal{N}(0, I)\big), \tag{2}$$

where $\beta$ balances reconstruction fidelity against latent regularization. By temporally downsampling the input token, the ST-VAE produces a sequence of latent vectors that both compress redundant frames and encode the underlying motion dynamics [17], yielding a compact token stream that supports more efficient and accurate downstream prediction.

## 3.4 Autoregressive Masked Generation with Wavelet Guidance

**Wavelet Feature Extraction.** Prior motion prediction works typically employ the Discrete Cosine Transform (DCT) for frequency-domain processing of motion. While DCT captures the low-frequency

motion trends, it overlooks high-frequency dynamics critical for subtle or rapid movements. To avoid this limitation, we employ the Discrete Wavelet Transform (DWT), which yields both low- and high-frequency subbands by contrast, furnishing richer spectral cues and improving prediction accuracy. To inject these spectral cues into our autoregressive diffusion backbone, we apply vanilla DWT operation to the raw 3D–joint history sequence along both temporal and channel axes. The transform uses a pair of $\ell$-length filters: a low-pass filter and a high-pass filter derived from a chosen discrete wavelet basis such as Harr. Specifically, given a motion sequence $x[i, j]$ $(i \in [1, H + F], j \in [1, 3J])$ and $a, b \in \{L, H\}$, we compute four subbands $(Y_{L,L}, Y_{L,H}, Y_{H,L}, Y_{H,H})$ as follows:

$$Y_{a,b}[k_1, k_2] = \sum_{i=1}^{H+F} \sum_{j=1}^{3J} f_a(i - 2k_1) f_b(j - 2k_2) x[i, j], \tag{3}$$

where $f_L$ and $f_H$ represent the low-pass and high-pass filters, respectively. Here, each subband $Y_{a,b} \in \mathbb{R}^{K \times D}$, $K = \lfloor \frac{H+F+\ell-1}{2} \rfloor$, $D = \lfloor \frac{3J+\ell-1}{2} \rfloor$, $k_1 \in [1, K]$, $k_2 \in [1, D]$. Concretely, $Y_{L,L}$ captures the low-frequency coefficients, $Y_{L,H}, Y_{H,L}$ capture the temporal- or spatial-detail coefficients, and $Y_{H,H}$ captures high-frequency detail in both axes. Then we concatenate the four subbands along the channel axis to get $Y = [Y_{L,L}, Y_{L,H}, Y_{H,L}, Y_{H,H}] \in \mathbb{R}^{K \times 4D}$ following the approach in [12]. Finally, we use a learned linear projection to map these 4D-dimensional features into the same $d$-dimensional latent space as the ST-VAE tokens as follows:

$$F_{\text{wave}}[b] = \text{LN}(W Y + b) \in \mathbb{R}^{K \times D}, \tag{4}$$

where LN denotes the layer normalization and $W$ is the learned linear matrix.

By explicitly computing both approximation and detail coefficients, the processed wavelet subbands provide the later Transformer fusion block with direct access to multi-scale, time-frequency motion patterns—enabling more precise recovery of masked future tokens that better align with the input historical information.

**Wavelet-guided Masked Fusion Module.** To organically fuse spectral cues with mask embeddings for more precise modeling of the original motion, during training, the original motion input is first processed by ST-VAE to produce continuous-valued tokens. We first mask out a random subset of the tokens with a learnable [MASK] embedding to obtain $\mathbf{X}_{\text{masked}} \in \mathbb{R}^{T' \times d}$. We then add a shared positional encoding $\mathbf{E}_{\text{pos}} \in \mathbb{R}^{T' \times d}$ to form the initial token stream.

In parallel, the wavelet branch produces frequency-domain features $F_{\text{wave}} \in \mathbb{R}^{K \times d}$. We then design a fused Transformer to learn interactions between frequency-domain features and time-domain motion tokens. Specifically, we partition the Transformer stack into two phases—early local fusion with cross-attention, followed by later global aggregation via self-attention. 1) The first $N_{\text{local}}$ Transformer blocks are **local fusion layers**, each consisting of a cross-attention sublayer followed immediately by a self-attention sublayer. At layer $\ell = 1, \ldots, N_{\text{local}}$, these local layers tightly fuse time-domain tokens with frequency cues via cross-attention before letting them attend globally to one another. 2) After the $N_{\text{local}}$ local fusion layers, the remaining $L - N_{\text{local}}$ Transformer blocks act as **global context layers**, each containing only a standard self-attention sublayer plus feed-forward network—no further cross-attention to the wavelet branch. This mechanism enables the model to first inject complementary frequency information into the masked latents, and then to refine the joint representation through additional layers of purely temporal self-attention before projecting back to the diffusion MLPs. Specially, for local fusion layers, we first calculate the fusion of frequency features and latent features:

$$X' = \text{CrossAttn}(Q, K, V) = \text{softmax}\left(\frac{(QW_Q) \cdot (KW_K)^T}{\sqrt{d_k}}\right)(VW_V),$$

$$X_{out} = \text{SelfAttn}(Q, K, V) = \text{softmax}\left(\frac{(X'W'_Q) \cdot (X'W_K')^T}{\sqrt{d_k}}\right)(X'W_V'), \tag{5}$$

where $W_Q, W'_Q, W_K, W'_K, W'_V$ and $W_V$ are trainable weight matrices. $d_k$ is the dimensionality of the key vector $K$. The query $Q$ comes from the latent representations, while the keys $K$ and values $V$ are derived from the frequency-domain features. For the rest layers, the $l_{th}$ layer's result is calculated as follows:

$$X^{(l)} = \text{SelfAttn}(Q, K, V) = \text{softmax}\left(\frac{(X^{(l-1)}W_Q) \cdot (X^{(l-1)}W_K)^T}{\sqrt{d_k}}\right)(X^{(l-1)}W_V) \tag{6}$$

where $X^{(l-1)}$ is the output of previous layer.

**Diffusion for Autoregressive Prediction.** As shown in MAR [29], relying solely on autoregressive model (without diffusion prediction head) cannot capture chained token dependencies, resulting in poor generation quality. Inspired by its success in image synthesis, we incorporate the MAR diffusion loss into our human motion prediction framework. Given the fused representation $X^{(L)}$ of masked latents and wavelet embeddings, we perform autoregressive diffusion to reconstruct the original motion tokens. Instead of modeling the full joint distribution in a single pass, we generate each token conditioned on all previously generated tokens, following the continuous-valued MAR paradigm adapted to motion data:

$$p(x_1, \ldots, x_N) = \prod_{i=1}^{N} p(x_i \mid x_{1:i-1}), \tag{7}$$

where $\{x_i\}_{i=1}^{n}$ is the sequence of tokens, and the index $i$ runs from 1 to $n$, specifying their order.

We use our wavelet-guided fused Transformer to produce, for each position $i$, a conditioning vector $z_i = f(X_{1:i-1}^{(L)}; \mathrm{DWT}) \in \mathbb{R}^d$ for the diffusion network in MAR. To learn $p(x_i \mid z_i)$ in continuous space, we corrupt the ground-truth token $x_i$ under the standard DDPM schedule and train a noise predictor to recover the added noise via a diffusion loss objective:

$$x_i^{(t)} = \bar{\alpha}_t x_i + (1 - \bar{\alpha}_t)\,\epsilon, \quad \epsilon \sim \mathcal{N}(0, I), \; t = 1, \ldots, T. \tag{8}$$

$$\mathcal{L}_{\mathrm{diff}} = \mathbb{E}_{i,t,\epsilon}\big[\|\epsilon - \epsilon_\theta(x_i^{(t)} \mid t, z_i)\|_2^2\big]. \tag{9}$$

where $\bar{\alpha}_t$ is a noise schedule indexed by time step $t$, and $\epsilon_\theta(x_i^{(t)} \mid t, z_i)$ denotes the neural network's prediction of the noise, taking $x_i^{(t)}$ as input and being conditioned on both the time $t$ and the conditioning variable $z_i$. We implement $\epsilon_\theta$ as a three-block MLP with residual connections and LayerNorm.

## 3.5 Inference

At test time, we first encode the observed history $\mathbf{x}_{1:H} \in \mathbb{R}^{H \times 3J}$ with the ST-VAE encoder to obtain continuous latents $\mathbf{X}'_{1:H} = \mathrm{Enc}(\mathbf{x}_{1:H})$. We then form a full-length token sequence $\mathbf{u}^{(0)} \in \mathbb{R}^{(H+F) \times d}$ by setting its first $H$ rows to $\mathbf{z}_{1:H}$ and masking all $F$ future positions with a learnable [MASK] embedding. Over $K$ autoregressive iterations, at autoregressive step $k$ we compute the unmasking ratio $\rho_k = \cos\left(\frac{\pi k}{2K}\right)$ and select $\rho_k F$ of the still-masked tokens to predict in this round. We feed the current $\mathbf{u}^{(k-1)}$ together with the precomputed wavelet embeddings $\mathrm{DWT}(\mathbf{x}_{1:H})$ into the diffusion network $f_\theta$. Only the newly unmasked positions $i$ are updated by denoising sampler at diffusion time step $t$:

$$u_i^{(k)} = \frac{1}{\sqrt{\alpha_t}}\left(u_i^{(k-1)} - \frac{1 - \alpha_t}{\sqrt{1 - \bar{\alpha}_t}}\,\epsilon_\theta\big(u_i^{(k-1)} \mid t, z_i\big)\right) + \sigma_t \epsilon, \quad \epsilon \sim \mathcal{N}(0, I), \tag{10}$$

where $k = 1, \ldots, K$ indexes the autoregressive iterations, and $t = 1, \ldots, T$ indexes the diffusion timestep. Here, $\alpha_t$, $\bar{\alpha}_t$, and $\sigma_t$ are the diffusion noise schedule parameters at diffusion step $t$, and $\epsilon_\theta(\cdot \mid t, z)$ is the noise predictor conditioned on latents $z$. while all other tokens remain fixed. After $T$ rounds, every future token has been filled in. Finally, we extract $\mathbf{u}_{H+1:H+F}^{(T)}$ and decode it via the ST-VAE decoder to get the final prediction result:

$$\hat{\mathbf{x}}_{H+1:H+F} = \mathrm{Dec}\big(\mathbf{u}_{H+1:H+F}^{(T)}\big) \in \mathbb{R}^{F \times 3J}. \tag{11}$$

## 4 Experiments

### 4.1 Experimental Settings

**Datasets.** We evaluate our method on two widely adopted benchmarks for stochastic human motion prediction (SHMP): Human3.6M [20], HumanEva-I [42] and AMASS[33]. Human3.6M is a large-scale benchmark containing 3.6 million frames of 3D human joint positions captured at 50 Hz, featuring seven actors performing 15 diverse activities (e.g., walking, eating, and discussing). To

Table 1: Quantitative comparison on HumanEva-I and Human3.6M.

| Method | HumanEva-I | | | | | Human3.6M | | | | |
|---|---|---|---|---|---|---|---|---|---|---|
| | APD↑ | ADE↓ | FDE↓ | MMADE↓ | MMFDE↓ | APD↑ | ADE↓ | FDE↓ | MMADE↓ | MMFDE↓ |
| DeLiGAN[19] | 2.177 | 0.306 | 0.322 | 0.385 | 0.371 | 6.509 | 0.483 | 0.534 | 0.520 | 0.545 |
| DSF[57] | 4.538 | 0.273 | 0.290 | 0.364 | 0.340 | 9.330 | 0.493 | 0.592 | 0.550 | 0.599 |
| BoM [4] | 2.846 | 0.271 | 0.279 | 0.373 | 0.351 | 6.265 | 0.448 | 0.533 | 0.514 | 0.544 |
| DLow[58] | 4.855 | 0.251 | 0.268 | 0.362 | 0.339 | 11.741 | 0.425 | 0.518 | 0.495 | 0.531 |
| GSPS[35] | 5.825 | 0.233 | 0.244 | 0.343 | 0.331 | 14.757 | 0.389 | 0.496 | 0.476 | 0.525 |
| DivSamp[9] | 6.109 | 0.220 | 0.234 | 0.342 | **0.316** | 15.310 | 0.370 | 0.485 | 0.475 | 0.516 |
| STARS[52] | 6.031 | 0.217 | 0.241 | **0.328** | 0.321 | **15.884** | 0.358 | 0.445 | **0.442** | **0.471** |
| MotionDiff[50] | 5.931 | 0.232 | 0.236 | 0.352 | 0.320 | 15.353 | 0.411 | 0.509 | 0.508 | 0.536 |
| BeLFusion[2] | – | – | – | – | – | 7.602 | 0.372 | 0.474 | 0.473 | 0.507 |
| HumanMAC[6] | **6.554** | 0.209 | 0.223 | 0.342 | 0.320 | 6.301 | 0.369 | 0.480 | 0.509 | 0.545 |
| CoMusion[43] | – | – | – | – | – | 7.632 | 0.350 | 0.458 | 0.494 | 0.506 |
| Ours | 3.128 | **0.199** | **0.201** | 0.354 | 0.337 | 4.458 | **0.347** | **0.452** | 0.513 | 0.535 |

ensure compatibility with prior studies, we follow the protocol of [6, 43], modeling each pose with a 16-joint skeleton. Given the first 0.5 seconds (25 frames) of observed motion, the task is to forecast the subsequent 2 seconds (100 frames). HumanEva-I provides 3D motion captured at 60 Hz from three actors each performing five distinct movements, with poses encoded as 15-joint skeletons. Following common practice, we use the first 0.25 s (15 frames) of each sequence as input and task our model with forecasting the next 1 s (60 frames) of motion. Results on AMASS dataset can be seen in supplementary materials.

**Baselines.** We compare our method against eleven representative baselines, covering VAE-based, GAN-based, and diffusion-based approaches: DeLiGAN, DSF, BoM, DLow, GSPS, DivSamp, STARS, MotionDiff, BeLFusion, HumanMAC, and CoMusion. Belfusion and Comusion didn't perform experiments on HumanEva-I dataset.

**Metrics.** We evaluate our model using five established metrics for stochastic human motion prediction. The *Average Pairwise Distance (APD)* quantifies diversity by computing the mean L2 distance between all generated motion samples. *Average Displacement Error (ADE)* measures overall sequence accuracy as the minimum average L2 distance between predictions and the ground truth, while *Final Displacement Error (FDE)* focuses on precision at the final predicted frame. To address multi-modal scenarios, *Multi-Modal ADE (MMADE)* and *Multi-Modal FDE (MMFDE)* extend these metrics by grouping ground truth sequences based on similar initial observations, ensuring robust evaluation of plausible diverse outcomes. These metrics collectively assess accuracy, temporal consistency, and diversity, critical for real-world applications requiring both precision and variability.

## 4.2 Implementation Details

We employ a lightweight ST-VAE with a two-layer encoder-decoder architecture. Each hidden layer has a dimension of 128, and we apply a temporal downsampling rate of 2. It is trained for 500 epochs with a batch size of 128. Spectral cues are injected by applying a vanilla Haar wavelet transform to the raw 3D-joint history. For the Human3.6M dataset, the diffusion backbone consists of 12 Transformer layers: the first 6 layers each combine self-attention and cross-attention over the wavelet embeddings, while the remaining 6 layers use only self-attention. We set the latent dimension to 256. For HumanEva-I, we use the same overall design but employ only 3 layers with both self- and cross-attention, followed by 3 self-attention layers, also with a latent dimension of 256. The noise prediction network in the diffusion model is a 3-layer MLP with a hidden dimension of 1024. We optimize the model for 200 epochs using the AdamW optimizer [32] with $\beta_1 = 0.5$, $\beta_2 = 0.99$, and an initial learning rate of $2 \times 10^{-4}$. A multi-step learning-rate scheduler with decay factor $\gamma = 0.9$ is applied, and the batch size is increased to 256 to stabilize training.

## 4.3 Comparison with the State-of-the-Arts

**Quantitative Comparison.** Table 1 reports quantitative results on HumanEva-I and Human3.6M, comparing our method against recent baselines. It can be seen that our approach achieves the best ADE and FDE values on both the HumanEva-I and Human3.6M datasets, indicating that our method provides the most accurate predictions. While our method yields a lower APD, we note that many

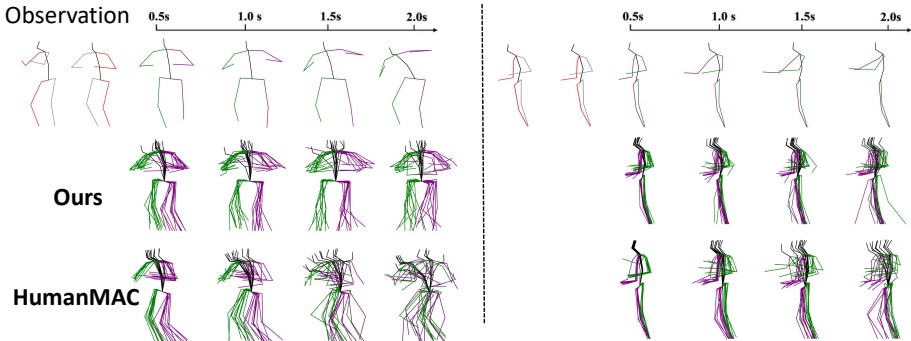

Figure 2: Qualitative comparisons: The first line is input history and ground truth motion, both methods predict ten predictions based on the same input history.

high-diversity models tend to sacrifice accuracy by generating trajectories that deviate from the observed history or true motions. These methods prioritize variety at the expense of fidelity. In contrast, our approach deliberately emphasizes producing the most plausible sequences, resulting in highly accurate predictions with more concentrated samples. As a consequence, although our method shows lower diversity, also reflected in the slightly worse MMADE and MMFDE scores compared to some competitors, it maintains a stronger adherence to the dominant motion modes. This highlights the trade-off between diversity and accuracy, with our model prioritizing precision while still ensuring an appropriate level of diversity.

**Qualitative Comparison.**   To further illustrate the effectiveness of our approach, Figure 4 presents qualitative predictions on two different actions. In each sequence, our model faithfully reproduces the nuanced joint trajectories of the ground truth, accurately capturing both smooth cyclic motions (e.g., walking) and rapid transitions. Moreover, our predictions exhibit superior stability: in the first example, the HumanMAC baseline produces an abrupt motion jump at the end of its forecast, whereas our model maintains coherent and physically plausible dynamics throughout. These visualizations demonstrate that our continuous-latent, wavelet-conditioned diffusion framework not only excels in numerical accuracy but also delivers compellingly realistic motion prediction.

Table 2: Comparison of inference time of different model sizes.

| Model | Params (M) | Inference Time (s) | APD↑ | ADE↓ | FDE↓ |
|---|---|---|---|---|---|
| HumanMAC | 28.4 | 1.25 | 6.301 | 0.369 | 0.480 |
| WaveAR (tiny) | 26.8 | **0.21** | **4.952** | 0.354 | 0.465 |
| WaveAR (small) | 51.9 | 0.43 | 4.458 | 0.347 | 0.452 |
| WaveAR (base) | 86.5 | 0.65 | 4.633 | **0.342** | **0.450** |

**Efficiency and Computational Analysis**   To demonstrate the efficiency and scalability of WaveAR, we provide three variants of our model with approximately 27M, 52M, and 87M parameters, respectively. We observe that enlarging the model size improves prediction accuracy but comes at the cost of reduced diversity. We provide a comparison of parameters and inference time for WaveAR and HumanMAC. The results are shown in Tab2. Thanks to the reduced number of denoising steps, WaveAR performs inference substantially faster than HumanMAC, while maintaining high prediction accuracy and realistic motion quality.

## 4.4   Motion In-Betweening via Flexible Masking

Beyond pure forecasting, our flexible masking strategy enables natural motion in-betweening between two distinct action sequences. By masking out an intermediate span of latent tokens and unmasking them progressively, the model seamlessly "fills in" a transition that respects both the initial and target poses. This adaptability arises from our autoregressive diffusion framework's ability to treat any masked interval—whether at the end of a sequence or in the middle—as a generation problem

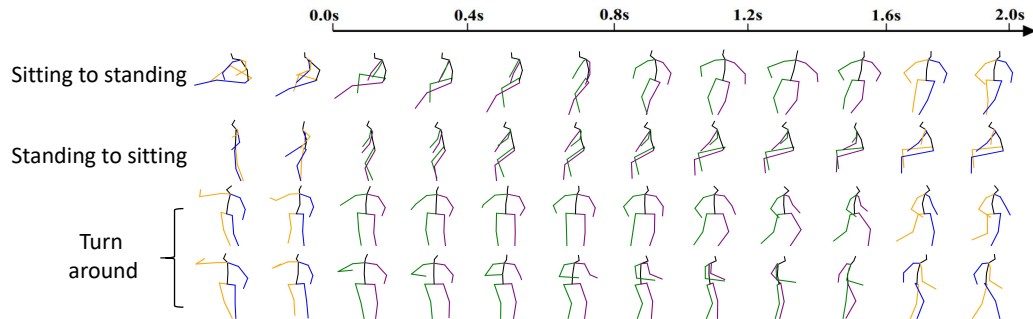

Figure 3: **Motion in-betweening results** of our proposed WaveAR model on the Human3.6M dataset. The first two columns represent the given initial motion, and the last two columns represent the target motion to be transitioned to. The visualization demonstrates that our model smoothly transitions from one motion to another. Both the initial and target motions consist of 20 frames.

Table 3: Ablation studies on proposed components, conducted on the Human3.6M dataset. Here, "w/o ST-VAE" indicates applying the downstream model directly on the raw input space; "w/ ST-VAE ($\gamma = 1$)" denotes an ST-VAE with a single downsampling (our implementation uses two downsamplings); "w/o DWT" removes the DWT branch.

| Method | APD↑ | ADE↓ | FDE↓ | MMADE↓ | MMFDE↓ |
|---|---|---|---|---|---|
| w/o ST-VAE | **8.625** | 0.492 | 0.641 | 0.570 | 0.692 |
| w/o DWT | 4.781 | 0.381 | 0.503 | 0.535 | 0.549 |
| w/ ST-VAE ($\gamma = 1$) | 5.263 | 0.446 | 0.586 | 0.564 | 0.597 |
| Ours | 4.458 | **0.347** | **0.451** | **0.513** | **0.535** |

conditioned on its surrounding context. The visualization results are presented in Figure 3. Our model smoothly transitions between motions. Each joint follows realistic physical constraints, ensuring the coherence and plausibility of the generated in-between frames. Not only can it handle simple motion transitions, such as turning, but it can also manage more complex sequences involving multiple action changes. For example, in the first row, the model successfully generates a sequence that transitions from a sitting posture to standing, followed by a turn and an exit motion. This demonstrates the model's ability to capture and generate dynamic transitions across varied motion types.

### 4.5 Ablation Study

In this section, we perform an ablation analysis on the Human3.6M dataset to investigate the impact of various design choices in our model, examining how each component contributes to its motion forecasting performance. Specifically, we conduct ablation experiments on the proposed frequency-domain module, the VAE module, the use of continuous latent vectors, and the diffusion settings. These experiments help us understand the individual contributions of each module and design choice, highlighting their influence on the overall performance of the model.

**Network Architecture Component.** We first conduct network component ablation experiments to evaluate the roles of different submodules in our architecture. The results are shown in Table 3.

**Diffusion for Autoregressive Prediction.** We next explore the contributions of diffusion-related components for our autoregressive prediction framework. First, we replace our continuous ST-VAE tokens with discrete codes from a VQ-VAE—this substitution leads to a marked degradation in both ADE and FDE, as shown in Table 4. Second, we remove the diffusion process entirely and instead use the output of the masked Transformer directly to reconstruct the original motion sequence via an $L_2$ loss; this baseline also performs poorly, confirming that the diffusion mechanism is essential for accurate, smooth motion generation.

**Wavelet-based Frequency Module**   We further analyze the impact of the proposed Discrete Wavelet Transform (DWT) module, which replaces the conventional Discrete Cosine Transform (DCT) used in prior frequency-based motion models. As shown in Table 5, introducing the DWT consistently improves motion prediction accuracy compared with both the baseline without any frequency module and the DCT-based variant. This demonstrates that the adaptive representation of wavelets better aligns with the temporal characteristics of motion sequences and provides more precise predictions.

Table 4: Performance comparison of different loss functions.

| Loss | APD ↑ | ADE ↓ | FDE ↓ | MMADE ↓ | MMFDE ↓ |
|------|-------|-------|-------|---------|---------|
| VQ-VAE | **7.233** | 0.477 | 0.589 | 0.544 | 0.631 |
| L2 Loss | 5.330 | 0.422 | 0.548 | 0.539 | 0.581 |
| Ours | 4.458 | **0.347** | **0.451** | **0.513** | **0.535** |

Table 5: Ablation study on different frequency-domain designs.

| Method | APD ↑ | ADE ↓ | FDE ↓ | MMADE ↓ | MMFDE ↓ |
|--------|-------|-------|-------|---------|---------|
| w/o DWT & DCT | 4.781 | 0.381 | 0.503 | 0.535 | 0.549 |
| w/ DCT | **5.016** | 0.362 | 0.478 | 0.521 | 0.537 |
| w/ DWT (Ours) | 4.458 | **0.347** | **0.451** | **0.513** | **0.535** |

## 5   Conclusion

In this paper, we propose WaveAR, a novel AR based framework, which is the first successful application of a continuous autoregressive generation paradigm to human motion prediction. Unlike prior methods that rely on discrete VQ-VAE codes, WaveAR employs a lightweight Spatio-Temporal VAE to encode raw 3D-joint sequences into smooth, quantization-free latents. Complementary spectral cues are injected via a 2D discrete wavelet transform and fused into the latent stream through local cross-attention layers, after which purely temporal self-attention layers refine the joint representation. A masked autoregressive diffusion process then generates each new token conditioned on its predecessors and rich wavelet features, yielding highly accurate, physically plausible motion forecasts. Extensive experiments on HumanEva-I and Human3.6M show that WaveAR achieves the lowest ADE and FDE among all baselines, confirming its superior precision.

## 6   Limitations

Our masked autoregressive diffusion module is primarily optimized to minimize reconstruction error over the average trajectory, which can lead to conservative or "safe" predictions, with diversity confined to a narrow range of motion variations.Besides, our current framework relies on standard joint-coordinate datasets containing only about 17–32 keypoints, which restricts its ability to capture subtle or fine-grained body movements. Addressing these limitations will be an important direction for our future work.

## Acknowledgements

This work was supported by National Natural Science Foundation of China (No. 62302297), the Fundamental Research Funds for the Central Universities (YG2023QNB17, YG2024QNA44), National Key R&D Program of China (2024YFE0115500), National Natural Science Foundation of China (No, 72192821, 62272447, 62472282, 62472285), Young Elite Scientists Sponsorship Program by CAST (2022QNRC001), Beijing Natural Science Foundation (L222117). We thank our project lead, Yabiao Wang, for his guidance.

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

## Supplementary Materials

## A    Per-Class Performance Comparison on Human3.6M

We conduct extensive experiments on the Human3.6M dataset, which encompasses 13 diverse action categories including Directions, Discussion, Eating, Greeting, Phoning, Photo, Posing, Purchases, Sitting, SittingDown, Smoking, Waiting, Walking, WalkDog, and WalkTogether. Table 6 presents a detailed comparison of our method against several state-of-the-art approaches across all action categories.From Table 6, it can be observed that our method demonstrates strong performance in terms of both ADE and FDE across a wide range of action categories. In particular, our approach achieves significant improvements on two walking-related actions — Walking and WalkTogether — where precise motion prediction is crucial. This indicates the model's enhanced ability to capture and forecast dynamic motion patterns, especially in scenarios involving coordinated or continuous movement.

## B    Wavelet guided masked fusion module

The detailed structure of our proposed Wavelet-guided masked fusion module is shown in Fig 4. The module consists of a hierarchical arrangement of two distinct layer types. The lower section comprises $N$ local fusion layers, each built with a sophisticated architecture incorporating CrossAttn, SelfAttn, and FFN components. Within these local fusion layers, the cross-attention mechanism facilitates information exchange between frequency domain features and masked vectors in the latent space, enabling effective multi-modal integration at a fine-grained level. The upper section contains $N$ global context layers, constructed exclusively with self-attention mechanisms and feed-forward networks. These layers are specifically designed to consolidate global information, progressively refining representations to produce a condition vector $z$ that encapsulates comprehensive contextual understanding. In our implementation, we set $N = 6$, creating a balanced architecture with sufficient capacity for both local feature fusion and global context integration.

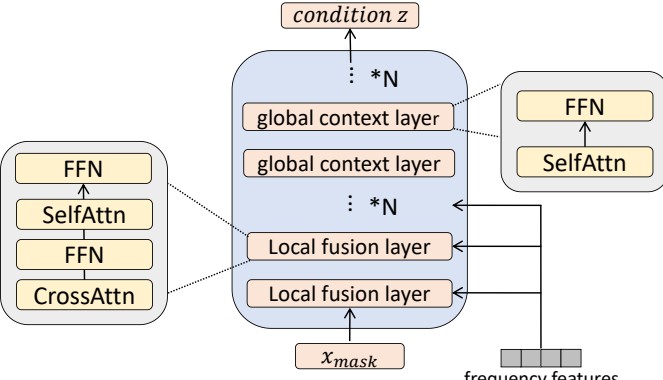

Figure 4: The detailed architecture of the Wavelet guided masked fusion module

## C    More ablation study

In this part, we conduct additional ablation studies to further analyze the effectiveness of key components in our method. Specifically, we examine: (1) the impact of varying the number of layers in the masked fusion module, and (2) the design of the denoising steps within the diffusion process.For the first ablation study, we investigate how different configurations of local fusion layers and total layers affect model performance. As shown in Table 7, we systematically vary the number of local fusion layers (ranging from 4 to 12) while adjusting the total number of layers (ranging from 8 to 14). The results demonstrate that a proper configuration of these architectural components significantly impacts model performance. Specifically, the model achieves optimal performance on our task when using 6 local fusion layers within a total of 12 layers, yielding the best overall performance. For the second ablation study, we examine different noise schedule configurations on

| Classes | | APD | ADE | FDE | MMADE | MMFDE | Classes | | APD | ADE | FDE | MMADE | MMFDE |
|---|---|---|---|---|---|---|---|---|---|---|---|---|---|
| Directions | TPK | 6.510 | 0.447 | 0.482 | 0.523 | 0.544 | Sitting | TPK | 6.417 | 0.400 | 0.547 | 0.461 | 0.548 |
| | DLow | 11.874 | 0.415 | 0.465 | 0.499 | 0.514 | | DLow | 11.425 | 0.364 | 0.605 | 0.440 | 0.523 |
| | GSPS | 15.398 | 0.407 | 0.477 | 0.492 | 0.522 | | GSPS | 14.966 | 0.323 | 0.454 | 0.411 | 0.484 |
| | DivSamp | **15.663** | 0.389 | 0.463 | 0.502 | 0.523 | | DivSamp | **15.614** | 0.317 | 0.465 | 0.417 | 0.490 |
| | BelFusion | 7.090 | 0.378 | 0.422 | 0.484 | 0.494 | | BelFusion | 6.495 | _0.306_ | _0.446_ | **0.400** | **0.461** |
| | HumanMAC | 6.357 | 0.391 | 0.456 | 0.475 | 0.475 | | HumanMAC | 5.941 | 0.312 | 0.456 | _0.404_ | 0.472 |
| | CoMusion | 7.527 | _0.372_ | **0.417** | **0.454** | **0.435** | | CoMusion | 6.237 | 0.307 | 0.448 | 0.406 | _0.468_ |
| | Ours | 4.673 | **0.371** | _0.421_ | _0.467_ | _0.461_ | | Ours | 4.080 | **0.302** | **0.443** | 0.437 | 0.492 |
| Discussion | TPK | 6.966 | 0.511 | 0.581 | 0.570 | 0.600 | SittingDown | TPK | 7.393 | 0.496 | 0.678 | 0.531 | 0.666 |
| | DLow | 11.872 | 0.472 | 0.536 | 0.533 | 0.549 | | DLow | 12.044 | 0.451 | 0.605 | 0.495 | 0.606 |
| | GSPS | 14.099 | 0.448 | 0.541 | _0.526_ | 0.563 | | GSPS | 13.725 | 0.406 | 0.561 | **0.461** | **0.565** |
| | DivSamp | 15.310 | 0.432 | 0.526 | 0.534 | 0.557 | | DivSamp | **14.899** | 0.413 | 0.579 | 0.478 | 0.586 |
| | BelFusion | 9.172 | 0.420 | 0.507 | **0.512** | _0.530_ | | BelFusion | 9.026 | 0.413 | 0.585 | _0.468_ | 0.587 |
| | HumanMAC | 7.496 | 0.434 | 0.533 | 0.547 | 0.571 | | HumanMAC | 6.871 | _0.381_ | **0.530** | 0.471 | _0.568_ |
| | CoMusion | 8.747 | _0.409_ | _0.497_ | 0.527 | **0.523** | | CoMusion | 7.253 | **0.378** | _0.546_ | 0.472 | 0.578 |
| | Ours | 5.420 | **0.402** | **0.489** | 0.534 | 0.541 | | Ours | 4.993 | 0.395 | 0.581 | 0.504 | 0.589 |
| Eating | TPK | 6.412 | 0.388 | 0.473 | 0.452 | 0.472 | Smoking | TPK | 6.522 | 0.422 | 0.529 | 0.509 | 0.560 |
| | DLow | 11.603 | 0.358 | 0.433 | 0.439 | 0.452 | | DLow | 11.549 | 0.400 | 0.515 | 0.490 | 0.537 |
| | GSPS | 15.570 | 0.334 | 0.419 | 0.424 | 0.448 | | GSPS | 14.822 | 0.466 | 0.485 | 0.472 | 0.530 |
| | DivSamp | _15.681_ | 0.321 | 0.419 | 0.428 | 0.445 | | DivSamp | **15.688** | 0.353 | 0.486 | 0.475 | 0.523 |
| | BelFusion | 5.954 | 0.310 | 0.381 | 0.418 | 0.420 | | BelFusion | 6.780 | 0.341 | 0.467 | 0.467 | 0.512 |
| | HumanMAC | 4.817 | 0.305 | 0.374 | _0.411_ | _0.409_ | | HumanMAC | 5.415 | 0.339 | 0.475 | _0.445_ | 0.501 |
| | CoMusion | 6.149 | **0.295** | **0.366** | **0.408** | **0.395** | | CoMusion | 6.802 | **0.311** | **0.443** | **0.427** | **0.458** |
| | Ours | 3.461 | _0.303_ | _0.371_ | 0.420 | 0.413 | | Ours | 4.222 | _0.322_ | _0.447_ | 0.451 | _0.486_ |
| Greeting | TPK | 6.779 | 0.555 | 0.615 | 0.571 | 0.598 | Waiting | TPK | 6.631 | 0.480 | 0.584 | 0.526 | 0.568 |
| | DLow | 11.897 | 0.530 | 0.590 | 0.561 | 0.564 | | DLow | 11.680 | 0.441 | 0.541 | 0.497 | 0.534 |
| | GSPS | 14.974 | 0.502 | 0.592 | _0.532_ | 0.577 | | GSPS | 15.000 | 0.400 | 0.514 | _0.475_ | 0.529 |
| | DivSamp | 15.447 | 0.489 | 0.575 | 0.535 | 0.562 | | DivSamp | 15.455 | 0.387 | 0.517 | 0.486 | 0.535 |
| | BelFusion | 8.482 | 0.482 | **0.544** | **0.524** | **0.540** | | BelFusion | 7.747 | 0.390 | 0.507 | **0.471** | _0.511_ |
| | HumanMAC | 7.939 | 0.499 | 0.571 | 0.573 | 0.592 | | HumanMAC | 6.506 | 0.385 | 0.532 | 0.496 | 0.557 |
| | CoMusion | 8.946 | _0.481_ | 0.556 | 0.558 | 0.552 | | CoMusion | 7.690 | **0.358** | **0.484** | 0.487 | **0.476** |
| | Ours | 5.444 | **0.473** | _0.551_ | 0.554 | _0.551_ | | Ours | 4.434 | _0.368_ | _0.506_ | 0.515 | 0.538 |
| Phoning | TPK | 6.410 | 0.377 | 0.475 | 0.468 | 0.507 | WalkDog | TPK | 7.384 | 0.560 | 0.694 | 0.592 | 0.665 |
| | DLow | 11.542 | 0.343 | 0.444 | 0.451 | 0.487 | | DLow | 11.882 | 0.490 | 0.566 | 0.539 | 0.570 |
| | GSPS | 15.050 | 0.311 | 0.413 | 0.436 | 0.476 | | GSPS | 13.746 | 0.459 | 0.564 | 0.530 | 0.587 |
| | DivSamp | 15.751 | 0.296 | 0.400 | 0.437 | 0.471 | | DivSamp | **15.616** | 0.439 | 0.555 | 0.532 | 0.577 |
| | BelFusion | 6.649 | 0.283 | 0.375 | 0.426 | 0.445 | | BelFusion | 9.335 | 0.432 | **0.530** | _0.527_ | _0.569_ |
| | HumanMAC | 5.069 | 0.287 | 0.383 | 0.405 | 0.431 | | HumanMAC | 7.741 | 0.441 | 0.540 | 0.543 | 0.591 |
| | CoMusion | 6.427 | _0.268_ | **0.363** | **0.390** | **0.399** | | CoMusion | 9.154 | **0.426** | 0.540 | **0.520** | **0.554** |
| | Ours | 4.013 | **0.264** | **0.363** | _0.404_ | _0.421_ | | Ours | 5.823 | _0.431_ | _0.536_ | 0.566 | 0.601 |
| Photo | TPK | 6.894 | 0.541 | 0.689 | 0.548 | 0.633 | WalkTogether | TPK | 6.718 | 0.443 | 0.548 | 0.535 | 0.573 |
| | DLow | 11.931 | 0.507 | 0.655 | 0.516 | 0.596 | | DLow | 11.951 | 0.395 | 0.495 | 0.503 | 0.530 |
| | GSPS | 14.310 | 0.485 | 0.663 | _0.502_ | 0.606 | | GSPS | 15.030 | 0.316 | 0.440 | 0.473 | 0.516 |
| | DivSamp | 15.330 | 0.474 | 0.665 | 0.506 | 0.607 | | DivSamp | **16.095** | 0.321 | 0.458 | 0.486 | 0.525 |
| | BelFusion | 8.446 | _0.434_ | 0.601 | **0.462** | **0.546** | | BelFusion | 6.378 | 0.296 | 0.393 | 0.484 | 0.495 |
| | HumanMAC | 7.505 | 0.438 | _0.600_ | 0.511 | 0.619 | | HumanMAC | 4.336 | 0.298 | 0.387 | _0.447_ | 0.454 |
| | CoMusion | 8.923 | **0.422** | 0.606 | 0.503 | 0.611 | | CoMusion | 6.512 | _0.270_ | _0.372_ | **0.435** | **0.431** |
| | Ours | 5.522 | 0.445 | **0.591** | 0.531 | _0.590_ | | Ours | 4.128 | **0.256** | **0.357** | 0.449 | _0.451_ |
| Posing | TPK | 6.520 | 0.466 | 0.538 | 0.542 | 0.565 | Walking | TPK | 6.708 | 0.455 | 0.533 | 0.538 | 0.558 |
| | DLow | 11.875 | 0.442 | 0.521 | 0.510 | _0.525_ | | DLow | 11.904 | 0.428 | 0.518 | 0.516 | 0.539 |
| | GSPS | 15.149 | 0.415 | 0.527 | 0.498 | 0.543 | | GSPS | 14.797 | 0.351 | 0.469 | 0.490 | 0.528 |
| | DivSamp | **15.429** | 0.395 | 0.499 | 0.510 | 0.541 | | DivSamp | **15.964** | 0.373 | 0.535 | 0.508 | 0.547 |
| | BelFusion | 8.438 | 0.406 | 0.510 | _0.498_ | 0.531 | | BelFusion | 5.116 | 0.367 | 0.471 | 0.530 | 0.546 |
| | HumanMAC | 7.320 | 0.407 | 0.530 | 0.512 | 0.553 | | HumanMAC | 4.306 | 0.321 | 0.447 | 0.472 | 0.485 |
| | CoMusion | 8.236 | _0.393_ | _0.501_ | 0.492 | **0.499** | | CoMusion | 6.487 | _0.308_ | _0.443_ | **0.447** | **0.465** |
| | Ours | 5.143 | **0.389** | **0.497** | 0.532 | 0.537 | | Ours | 3.522 | **0.263** | **0.406** | _0.452_ | _0.474_ |
| Purchases | TPK | 7.450 | 0.505 | 0.522 | 0.535 | 0.538 | | | | | | | |
| | DLow | 11.947 | 0.430 | 0.422 | **0.493** | 0.477 | | | | | | | |
| | GSPS | 13.969 | 0.414 | 0.429 | 0.497 | 0.497 | | | | | | | |
| | DivSamp | **14.967** | 0.388 | **0.404** | 0.502 | 0.478 | | | | | | | |
| | BelFusion | 10.272 | 0.410 | _0.409_ | _0.494_ | 0.472 | | | | | | | |
| | HumanMAC | 8.601 | _0.403_ | 0.410 | 0.506 | _0.439_ | | | | | | | |
| | CoMusion | 9.484 | 0.405 | 0.426 | 0.496 | **0.425** | | | | | | | |
| | Ours | 5.185 | _0.403_ | 0.421 | 0.542 | 0.460 | | | | | | | |

Table 6: Comparison of different methods on various classes and metrics.

model performance and inference efficiency. Table8 shows that reducing both noising steps during training and DDIM steps during inference to 10 maintains competitive performance while drastically reducing computational costs compared to larger step configurations. This demonstrates our approach can maintain high prediction accuracy even with a significantly accelerated sampling process, making it more practical for real-time applications.

| local_layer | total_layer | APD↑ | ADE↓ | FDE↓ | MMADE↓ | MMFDE↓ |
|---|---|---|---|---|---|---|
| 4 | 8 | **5.013** | 0.372 | 0.484 | 0.529 | 0.542 |
| 5 | 10 | 4.766 | 0.361 | 0.467 | 0.515 | **0.533** |
| 6 | 12 | 4.458 | **0.347** | **0.452** | **0.513** | 0.535 |
| 12 | 12 | 4.229 | 0.354 | 0.461 | 0.520 | 0.536 |
| 7 | 14 | 4.587 | 0.362 | 0.458 | 0.522 | 0.540 |

Table 7: Performance comparison with different configurations of local fusion layers and total layers.

| Noising steps | DDIM steps | APD↑ | ADE↓ | FDE↓ | MMADE↓ | MMFDE↓ |
|---|---|---|---|---|---|---|
| 1000 | 100 | **5.172** | 0.350 | 0.454 | 0.518 | **0.533** |
| 100 | 10 | 4.574 | 0.363 | 468 | 0.520 | 0.536 |
| 10 | 10 | 4.458 | **0.347** | **0.452** | **0.513** | 0.535 |

Table 8: Experiment results of the ablation study on diffusion steps

# D   Experiments on AMASS dataset

In Table 9 ,we report quantitative results on AMASS dataset[33]. As shown below, our model still achieves the best accuracy metrics (ADE&FDE), demonstrating strong generalization capability.

| Model | APD↑ | ADE↓ | FDE↓ | MMADE↓ | MMFDE↓ |
|---|---|---|---|---|---|
| Belfusion | 9.376 | 0.513 | 0.560 | 0.569 | 0.585 |
| HumanMAC | 9.321 | 0.511 | 0.554 | 0.593 | 0.591 |
| CoMusion | **10.848** | 0.494 | 0.547 | **0.469** | **0.466** |
| Ours | 7.022 | **0.485** | **0.538** | 0.562 | 0.587 |

Table 9: Performance comparison across different models on AMASS dataset

