# OpenReview forum: "WaveAR: Wavelet-Aware Continuous Autoregressive Diffusion for Accurate Human Motion Prediction"
_NeurIPS.cc/2025/Conference — NeurIPS 2025 poster_

### Official Review · Reviewer_76BV · 2025-06-19

**Clarity:** 4
**Significance:** 3
**Originality:** 3
**Rating:** 5
**Confidence:** 3

**Summary:**

This paper presents a new approach for stochastic human motion prediction. The proposed approach presents multiple components:
- First, the motion is encoded as a sequence of continuous tokens using a spatio-temporal VAE (ST-VAE). In comparison  with the recent autoregressive approaches that rely on vector-quantized representations, this continuous latent representation allows for preserving more details.
- Second, predictions are made in the latent space in an autoregressive fashion, conditioned on wavelet features extracted from the motion. This step combines a masked generative approach with a diffusion process. The wavelet transforms preserve more fine-grained features than the commonly used DCT, while the designed generation process allows for realistic, diverse predictions.

Experiments on two widely used benchmarks demonstrate good performance compared to state-of-the-art approaches.

**Questions:**

In the rebuttal, I would like the authors to discuss the following points:
- What are the limitations of the proposed approach, and what could be future improvements over WaveAR?
- Ablations on the DCT vs. Wavelets
- Discussion on the difference between training and inference settings (past always fully visible at inference, wavelet features computed only on the past at inference).

**Ethical Concerns:**

["NO or VERY MINOR ethics concerns only"]

**Final Justification:**

Following the rebuttal and discussions, I still believe this paper should be accepted. The approach presented in this paper is well-explained and novel: this is the first application of continuous autoregressive modeling for human motion generation, and the integration of discrete wavelet transforms in the framework is a valuable improvement. The model is flexible (it can be used for motion generation and in-betweening) and obtains strong performance on various benchmarks.

In the final version, I believe it would be important to add: (1) Ablation on the DCT vs. Wavelets; (2) Discussion of the limitations and future works; (3) Experiments on the AMASS dataset; (4) Information about running time and efficiency.

Discussion/comparison with MotionWavelet would be valuable even if it is not published work.

**Limitations:**

Limitations and potential negative impacts are not mentioned at all (the checklist says it is in the supplementary materials, but I could not find it).

**Quality:**

3

**Strengths And Weaknesses:**

# Strengths
I feel like this paper has many strengths:
- The approach is very well motivated: the introduction shows the current limitations of prior works and introduces the contributions that will address them.
- The approach is novel: to my knowledge, this is the first that uses continuous autoregressive generation for human motion prediction.
- The method is clear and well-explained. Despite the many components of the approach (tokenization, wavelet features, autoregressive generation with masking, diffusion process), the paper still reads nicely and is very understandable.
- The masking allows the use of the same model for motion prediction and motion in-betweening.
- The experiments follow the common practice, and the method obtains good results compared to the state of the art (better accuracy and very decent diversity).
- Ablations are well-designed to demonstrate the interest of the technical contributions.
- Visualisations given in the supplementary materials show good qualitative performance, with accurate yet diverse outputs.

# Weaknesses

I do not find major weaknesses in this paper. However, here are some points that I think could still be improved:

- There is no discussion at all about the limitations of the method and future lines of research.
- Additional ablation studies could be performed. In particular, it would be interesting to see how the model would perform with DCT features instead of wavelets.
- The method section could be even clearer with improved notations. For instance, the symbol sigma is used for different purposes in Eqs. 1 and 10 (even if its meaning is clearly explained in both equations). L187, wouldn't it be clearer to use $L'$ instead of $T'$ as the sequence of $T$ poses is encoded to $L$ tokens before masking?
- I think there is a small mistake L227-228: The ST-VAE is supposed to reduce the temporal dimension, but here, the length of the input and the tokenized representation is the same (H).
- L145: The reference to the figure should be given (not fig. X).
- From my understanding, at training time, the wavelet features contain information about the entire sequence, including the tokens that are masked. Wouldn't it be better to have something more similar to the inference settings at training time, with wavelet features computed only on the past, and masking done solely on the future?

# Overall recommendation

This paper presents many strengths: the approach is well motivated and explained, novel, and obtains good results. Additionally, the ablations give insights into the most important parts of the model, and qualitative results are as impressive as quantitative ones. While I don't find major weaknesses in this paper, I feel that discussions on the current limitations and future works, as well as additional ablations, could improve it. I recommend "Accept".

---

> ### Author Rebuttal · Authors · 2025-07-31
>
> To Reviewer 76BV
> ===
>
> We sincerely appreciate Reviewer 76BV for the thoughtful review. Below, we provide detailed responses to each comment.
>
> **Q1：What are the limitations of the proposed approach, and what could be future improvements over WaveAR?**
>
> Thanks to the reviewer for the reminder. While WaveAR achieves strong average accuracy (ADE&FDE), it does exhibit two main limitations: Our masked autoregressive diffusion module focuses on minimizing reconstruction error over the average trajectory, which can lead to overly “safe” predictions, with diversity confined to a relatively small range of motion variations; Besides,current approach relies on standard joint‑coordinate datasets containing only about 20 keypoints, limiting its ability to capture finer movements such as hand motions .Enabling fine‑grained motion details is thus an exciting direction for future research.
>
> **Q2:Ablations on the DCT vs. Wavelets**
>
> From the table, we see that incorporating spectral information improves motion prediction accuracy, and that DWT—by capturing high-frequency details in the input beyond what DCT can—yields even more precise predictions.
>
> | |APD↑|ADE↓|FDE↓|MMADE↓|MMFDE↓|
> |:----|:----|:----|:----|:----|:----|
> |w/o DWT&DCT|4.781|0.381|0.503|0.535|0.549|
> |w/ DCT|**5.016**|0.362|0.478|0.521|0.537|
> |w/ DWT|4.458|**0.347**|**0.451**|**0.513**|**0.535**|
>
> **Q3:Differences between training and inference process**
>
> Thank you for your careful review and valuable suggestions. As stated in paper L167-L168 "To inject these spectral cues into our autoregressive diffusion backbone, we apply vanilla DWT operation to the raw 3D–joint **history sequence** along both temporal and channel axes." The wavelet features are computed only on the observed history frames of length H at both training and inference time.The upper limit H+F in Equation (3) merely denotes the length of the input sequence for the wavelet transform computation. This may cause ambiguity and will revise it in the next version for clarity. For the masking strategy, the masking applied to the latent sequence during training is performed randomly across the entire sequence according to a predefined probability, not limited to future prediction frames. This flexible masking strategy enables our model not only to perform standard autoregressive prediction of future frames from history but also to support more complex tasks such as motion switching.
> In summary, the wavelet features provide crucial spectral cues that enhance temporal understanding, and the autoregressive denoising diffusion model delivers strong generative power to ensure accurate and robust sequence reconstruction.
>
> **Q4:Minor typos**
>
> We thank the reviewer for  careful attention in identifying these typos and ambiguities; we will correct them in the final version.

---

> > ### Comment · Reviewer_76BV · 2025-08-01
> >
> > Thanks to the authors for this rebuttal, that effectively addresses the minor concerns I had. I just have a one more question related to the rebuttal:
> >
> > **Q1**: To address the limited diversity in the results, would it be possible to increase randomness at inference (for instance by increasing the variance in the diffusion step of Eq. 10)? This would probably decrease slightly the accuracy, but might help increasing diversity.

---

> > > ### Author Response · Authors · 2025-08-04
> > >
> > > Dear Reviewer 76BV:
> > >
> > > We greatly appreciate the time and effort you have devoted to reviewing our paper!
> > >
> > > It is true that scaling the diffusion variance can increase diversity. In practice, we have also tried several other approaches to boost diversity—such as increasing the temperature in the autoregressive inference step and reducing the ST-VAE compression ratio—all of these methods result in a higher diversity, but also lead to varying degrees of accuracy degradation. For example, the table below illustrates how adjusting the VAE scaling factor impacts prediction accuracy (This table is from Table 2 in the paper, you can refer to the paper for detailed information). Our current model achieves the highest accuracy while still preserving a certain degree of diversity, and crucially, it never produces results that violate the underlying physical laws. In future work, we will also explore strategies to achieve a better trade-off between diversity and accuracy.
> > >
> > > |Method|APD ↑|ADE ↓|FDE ↓|MMADE ↓|MMFDE ↓|
> > > |:----|:----|:----|:----|:----|:----|
> > > |ST-VAE (γ = 1)|**5.263**|0.446|0.586|0.564|0.597|
> > > |ST-VAE (γ = 2)(Ours)|4.458|**0.347**|**0.451**|**0.513**|**0.535**|

---

> > > > ### Comment · Reviewer_76BV · 2025-08-05
> > > >
> > > > Thank you for this detailed answer! I do not have further questions.

---

> > > > > ### Author Response · Authors · 2025-08-06
> > > > >
> > > > > Dear Reviewer 76BV:
> > > > >
> > > > > We are pleased that our response has addressed your concerns,Thank you again for your effort in the review and the discussion!
> > > > >
> > > > > Best regards,

---

### Official Review · Reviewer_p7mw · 2025-07-01

**Clarity:** 2
**Significance:** 2
**Originality:** 3
**Rating:** 4
**Confidence:** 5

**Summary:**

The paper presents WaveAR, a wavelet-aware autoregressive model for human motion prediction. The approach integrates frequency-domain features extracted via a discrete wavelet transform into masked latent representations using an attention-based fusion mechanism. A subsequent MLP module serves as a noise predictor within a diffusion process to estimate future motion sequences. The model is evaluated on the Human3.6M and HumanEva-I datasets and is compared against state-of-the-art methods using standard diversity and accuracy metrics.

**Questions:**

1. Could the authors provide a demonstration of the model’s robustness and generalizability on a recent, large-scale, and diverse dataset—such as AMASS, which has been adopted by several recent baselines?

2. The authors are encouraged to further analyze the trade-off between accuracy and diversity, and to discuss whether their model still preserves meaningful diversity within each mode, rather than collapsing to a limited set of dominant motions.

3. Could the authors provide more details or comparisons regarding the computational cost, inference speed, or model size of WaveAR relative to other baselines?

4. Minor typos: L.145

**Ethical Concerns:**

["NO or VERY MINOR ethics concerns only"]

**Final Justification:**

The rebuttal addressed some of my concerns regarding the benchmark evaluation and the trade-off between accuracy and diversity. I would like to raise my score to borderline accept.

**Limitations:**

The paper lacks a discussion of limitations. An analysis of issues such as the trade-off between accuracy and diversity, as well as potential failure cases, would contribute to a more balanced and transparent presentation of the work.

**Paper Formatting Concerns:**

The related work section primarily lists prior methods without clearly articulating how WaveAR differs from or improves upon them. A more detailed comparison highlighting the key distinctions and contributions relative to existing approaches would help better contextualize the novelty of the proposed method.

**Quality:**

3

**Strengths And Weaknesses:**

Strengths:

1. The paper includes thorough experiments including various ablations that analyze the contribution of each component offering insights into the model’s design choices.

2. The integration of wavelet transforms allows the model to capture motion information at multiple temporal scales, which can help in representing fine-grained motion patterns.



Weaknesses:

1. While the authors evaluate on standard benchmarks such as Human3.6M and HumanEva-I, recent works (e.g., BeLFusion, HumanMAC, and CoMusion) have demonstrated results on AMASS—a larger and more diverse dataset that better captures a wide range of human motions. The absence of evaluation on AMASS limits the assessment of the model’s robustness and weakens the strength of its generalizability claims.

2. While WaveAR emphasizes accurate motion prediction and achieves the best ADE and FDE scores, the improvement over previous methods is marginal and comes with a large drop in diversity, as indicated by significantly lower APD scores (e.g., compared to HumanMAC on HumanEva-I and CoMusion on Human3.6M). In the context of stochastic human motion prediction, diversity is critical to reflecting the multimodality of plausible future movements. For instance, in autonomous driving scenarios, a pedestrian turning right might vary subtly in trajectory or speed, and capturing these variations within each motion mode is crucial. This raises concerns about the model's ability to capture diverse and plausible future motions within each mode, which is essential for robust stochastic prediction.

3. Although the authors mention that WaveAR is computationally efficient and fast, they do not provide discussion of computational cost, inference speed, or model size relative to other baselines.

---

> ### Author Rebuttal · Authors · 2025-07-31
>
> To Reviewer p7mw
> ===
> We sincerely appreciate the valuable feedback from Reviewer p7mw. Our responses to each comment are provided below.
>
> **Q1: Results on AMASS dataset**
>
> Thank you for your suggestions. We conducted experiments on the AMASS dataset. As shown below, our model still achieves the best accuracy metrics (ADE&FDE), demonstrating strong generalization capability.
>
> | Model     | APD↑    | ADE↓   | FDE↓   | MMADE↓ | MMFDE↓ |
> |:--------- |:------ |:----- |:----- |:----- |:----- |
> | Belfusion | 9.376  | 0.513 | 0.560 | 0.569 | 0.585 |
> | HumanMAC  | 9.321  | 0.511 | 0.554 | 0.593 | 0.591 |
> | CoMusion  | **10.848** | 0.494 | 0.547 | **0.469** | **0.466** |
> | WaveAR(Ours)      | 7.022  | **0.485** | **0.538** | 0.562 | 0.587 |
>
> **Q2：Trade-off between accuracy and diversity**
>
> Thank you very much for your valuable suggestion regarding the analysis of accuracy-diversity trade-off. We acknowledge that our current model prioritizes prediction accuracy, resulting in relatively limited diversity. Previous methods have achieved higher diversity often at the expense of generating unrealistic or physically implausible motions  such as discontinuities in movement [1,2,3], an issue that does not arise in our proposed approach. To demonstrate the robustness and realism of our model predictions, we provide detailed results in the supplementary materials showing consistently accurate performance across various fine-grained action categories in the Human3.6M dataset. This further supports the validity and robustness of our method. Nevertheless, we recognize that the limited diversity remains a current limitation of our model. Improving diversity while preserving realism is an important future research direction. We will add this part in the Limitation section. In the future, we will pay more attention to this topic.
>
> Reference:
>
> [1]Mao, W., Liu, M., Salzmann, M.: Generating smooth pose sequences for diverse human motion prediction. In: ICCV. pp. 13289–13298 (2021)
>
>  [2]Dang, L., Nie, Y., Long, C., Zhang, Q., Li, G.: Diverse human motion prediction via gumbel-softmax sampling from an auxiliary space. In: MM. pp. 5162–5171 (2022)
>
> [3]Wei, D., Sun, H., Li, B., Lu, J., Li, W., Sun, X., Hu, S.: Human joint kinematics diffusion-refinement for stochastic motion prediction. In: AAAI. pp. 6110–6118 (2023)
>
> **Q3:Inference time and model size**
>
> Thank you for your suggestion. We evaluated inference time and parameter count on  NVIDIA V100 GPU by comparing three model variants of different sizes.
> | |Params|Inference time(s)|APD↑|ADE↓|FDE↓|
> |:----|:----|:----|:----|:----|:----|
> |HumanMAC|28.4M|1.25|**6.301**|0.369|0.480|
> |Ours(tiny)|26.8M|0.21|4.952|0.354|0.465|
> |Ours(small)|51.9M|0.43|4.458|0.347|0.452|
> |Ours(base)|86.5M|0.65|4.633|**0.342**|**0.450**|
>
> **Q4:Lack of Limmitations:**
>
> Thanks to the reviewer for the reminder. While WaveAR achieves strong average accuracy, it does exhibit two main limitations: Our masked autoregressive diffusion module focuses on minimizing reconstruction error over the average trajectory, which can lead to overly “safe” predictions, with diversity confined to a relatively small range of motion variations; Besides,current approach relies on standard joint‑coordinate datasets containing only about 17–32 keypoints, limiting its ability to capture finer movements .Enabling fine‑grained motion details is thus an exciting direction for future research.
>
> **Q5:Minor typos**
>
> Thank you for the reminder of minor typos, we will correct them in the final version.

---

> > ### Comment · Reviewer_p7mw · 2025-08-05
> > **Thank you for the rebuttal**
> >
> > Thank you for the detailed rebuttal, including the experiments on the suggested AMASS dataset and the discussion regarding the trade-off between accuracy and diversity. Some of my concerns have been addressed. I encourage the authors to include the additional results presented in the rebuttal, along with the expanded discussion of limitations, in the final version. I have no further questions at this time and will revisit my score following the discussion with the other reviewers.

---

> > > ### Author Response · Authors · 2025-08-07
> > >
> > > Dear Reviewer p7mw:
> > >
> > > Thank you for the time and effort you dedicated to reviewing our manuscript and for providing such valuable feedback. We are pleased to hear that our responses have addressed your concerns.  We  will add incorporate the additional experimental results into the final version of the paper and revise the wording in the related work section according to your suggestions.Thank you again for your insightful comments and suggestions.
> > >
> > > Best regards,

---

> ### Comment · Area_Chair_kPYJ · 2025-08-05
>
> Dear Reviewer p7mw,
>
> The deadline for author-reviewer discussion period is approaching. We kindly ask you to review the authors' rebuttal. Please provide your feedback soon. Thank you.
>
> Best,
>
> AC

---

### Official Review · Reviewer_fPmK · 2025-07-03

**Clarity:** 2
**Significance:** 2
**Originality:** 2
**Rating:** 3
**Confidence:** 4

**Summary:**

This paper presents WaveAR, a method for stochastic human motion prediction by autoregressively making predictions in a continuous latent space. By doing so, the paper claims to be avoiding potential performance degradations due to discretized tokens. Furthermore, this neural motion representation is further extended by Discrete Wavelet Transform (DWT) features that are intended to capture multi-scale low- and high-frequency motion details. The model first uses a Spatio-Temporal VAE to compress raw 3D joint sequences into a continuous latent token stream, eliminating quantization errors. Next, a masked autoregressive diffusion process predicts future tokens, by leveraging DWT features. The proposed approach is evaluated on Human3.6M and HumanEva-I benchmarks, demonstrating a better or comparable performance.

**Questions:**

See my weaknesses section for the main questions. I have one extra question:

1- (in line 3, abstract) “While autoregressive sequence models have excelled in related generation tasks, their reliance on vector-quantized tokenization limits motion fidelity and training stability.” Is there any evidence for this statement? The proposed ST-VAE can be similarly limited due to the inherent properties of the VAE formulation (e.g., reconstructing high-frequency details in the signal while ignoring the low-frequency ones).

**Ethical Concerns:**

["NO or VERY MINOR ethics concerns only"]

**Final Justification:**

The main takeaway from this paper is the improved accuracy-diversity trade-off. In certain applications, more accurate predictions are preferable to more diverse results. However, I have concerns regarding the novelty and contribution claims. While the lack of a quantitative comparison against MotionWavelet is not grounds for rejection, as it is an unpublished work, it was the first to present a wavelet representation for human motion. Similarly, the other contribution claim is being the first paper to apply "masked autoregressive modeling for human motion prediction." is not a novelty per se. Permutations of these techniques (e.g., discrete or continuous latent spaces, autoregressive modeling, masking, diffusion) have been used in previous motion modeling works (DartControl, MoMask, MLD), even if not specifically for motion prediction. Finally, the paper also requires a major revision regarding these contribution claims and the inclusion of additional AMASS results.

The reasons to reject—namely, the overstated novelty claims and the need for significant revisions—slightly outweigh the reasons to accept. Therefore, I will maintain my score of "Borderline Reject."

**Limitations:**

yes

**Quality:**

2

**Strengths And Weaknesses:**

**Strengths:**

1- Model Flexibility and Versatility: The model demonstrates significant flexibility. Beyond its primary function of forecasting, it can be effectively adapted for other tasks, such as motion in-betweening, by leveraging its flexible masking strategy.

2- Competitive Performance: The model performs better than or on-par with the baselines on standard benchmarks.

3- Integration of Techniques: A key strength lies in the novel adaptation of the MAR framework by guiding the autoregressive process with multi-scale features from a Discrete Wavelet Transform (DWT).

**Weaknesses:**

1- Limited Scope of Evaluation Benchmarks: The experimental evaluation is confined to the Human3.6M and HumanEva-I datasets. While these are standard benchmarks, the absence of tests on larger and more diverse datasets (e.g., CMU, AMASS) limits the assessment of the model's scalability and generalizability to more complex, in-the-wild motion.

2- Omission of a Key Baseline Comparison: The paper acknowledges that its use of DWT features is inspired by MotionWavelet, yet MotionWavelet is not included as a baseline in the quantitative comparisons. Given that MotionWavelet also achieves strong performance, it is difficult to determine the precise advantage of WaveAR's more complex architecture (which includes the ST-VAE, fusion module, and diffusion head) over a potentially simpler, wavelet-focused approach. This omission makes the contribution of the additional components less clear.

3- Potentially Overstated Novelty Claim: The central claim that WaveAR is "the first successful application of a continuous autoregressive generation paradigm to [the] SHMP domain" may be overstated. Earlier autoregressive models in human motion synthesis operated in continuous latent spaces before diffusion models became popular. For instance, HuMoR (Rempe et al., 2021) also predicts motion autoregressively in a continuous latent space. The paper's novelty claim should be refined to more accurately position the work within the existing literature, perhaps by emphasizing its combination of continuous autoregression with a wavelet-guided diffusion process.

---

> ### Author Rebuttal · Authors · 2025-07-31
>
> To Reviewer fPmk
> ==
>
> We sincerely appreciate Reviewer fPmk's thoughtful comment and insightful feedback. Please see our detailed response below.
>
> **Weakness (1):Limited Scope of Evaluation Benchmarks**
>
> Thank you for your suggestions. We conducted experiments on the AMASS dataset. As shown below, our model still achieves the best accuracy metrics (ADE&FDE), demonstrating strong generalization capability.
> | Model     | APD ↑   | ADE ↓  | FDE↓   | MMADE ↓| MMFDE↓ |
> |:--------- |:------ |:----- |:----- |:----- |:----- |
> | Belfusion | 9.376  | 0.513 | 0.560 | 0.569 | 0.585 |
> | HumanMAC  | 9.321  | 0.511 | 0.554 | 0.593 | 0.591 |
> | CoMusion  | **10.848** | 0.494 | 0.547 | **0.469** | **0.466** |
> | Ours      | 7.022  | **0.485** | **0.538** | 0.562 | 0.587 |
>
> **Weakness (2):Omission of a Key Baseline Comparison**
>
> Thank you for your suggestion. We compare MotionWavelet with our model and get the following results. FDE measures accuracy on the final frame, while ADE measures accuracy over the entire predicted sequence. Although MotionWavelet achieves a better FDE, its relatively poor ADE suggests it can produce inaccurate predictions during intermediate frames. Since ADE is a more important metric for overall trajectory accuracy, our method clearly outperforms MotionWavelet. Moreover, because MotionWavelet’s code is not publicly available, we re-implemented its algorithm based on the paper and compared inference speeds, demonstrating that our model runs significantly faster than MotionWavelet. We will add the results to the final version.
>
> | Model|APD↑|ADE↓|FDE↓ |Inference time(s)|
> |:----|:----|:----|:----|:----|
> |Motionwavelet|**6.506**|0.376|**0.408**|  1.65|
> |WaveAR(Ours)     | 5.016| **0.362**|0.478 | **0.43**|
>
> **Weakness (3). Potentially Overstated Novelty Claim**
>
> We appreciate the reviewer pointing out this issue. We agree that our original statement lacked precision and could lead to misunderstandings. To clarify, our approach is the first to apply masked autoregressive modeling for human motion prediction. We will correct our statement in the final version.
>
> **Questions (1). Evidence for the statement in line3**
>
> Autoregressive methods relying on VQ have achieved excellent performance on text-to-motion [1,2,3]. However, the presence of quantization error [2] and codebook collapse [4] makes it difficult to generalize these methods to other motion generation tasks. As noted in MARRS [5], VQ-VAE-based methods achieve poor performance for human action-reaction systhesis. For the human motion prediction, the results in Table 3 of our paper also demonstrate the instability of the VQ-VAE-based methods. For ease of reading, we have put the results from Table 3 in the paper into the rebuttal. When we replace our proposed ST-VAE with a VQ-VAE, the model’s performance degrades significantly across all metrics.
> |Loss|APD↑|ADE↓|FDE↓|MMADE↓|MMFDE↓|
> |:----|:----|:----|:----|:----|:----|
> |VQ-VAE|**7.233**|0.477|0.589|0.544|0.631|
> |WaveAR(Ours)|4.458|**0.347**|**0.451**|**0.513**|**0.535**|
>
>
> References:
>
> [1] Jianrong Zhang, Yangsong Zhang, Xiaodong Cun, Yong Zhang, Hongwei Zhao, Hongtao Lu, Xi Shen, and Ying Shan. Generating human motion from textual descriptions with discrete representations. In Proceedings of the IEEE/CVF conference on computer vision and pattern recognition, pages 14730–14740, 2023. 2, 3
>
> [2] Chuan Guo, Yuxuan Mu, Muhammad Gohar Javed, Sen Wang, and Li Cheng. Momask: Generative masked modeling of 3d human motions. In Proceedings of the IEEE/CVF Conference on Computer Vision and Pattern Recognition, pages 1900–1910, 2024. 1, 2, 4
>
> [3] kkasit Pinyoanuntapong, Pu Wang, Minwoo Lee, and Chen Chen. Mmm: Generative masked motion model. In Proceedings of the IEEE/CVF Conference on Computer Vision and Pattern Recognition, pages 1546–1555, 2024. 1, 2, 4, 8
>
> [4] huanxia Zheng and Andrea Vedaldi. Online clustered codebook. In Proceedings of the IEEE/CVF International Conference on Computer Vision, pages 22798–22807, 2023. 1
>
> [5] YB Wang, Shuo Wang, JN Zhang, JF Wu, QD He, CC Fu, CJ Wang, and Yong Liu. Marrs: Masked autoregressive unit-based reaction synthesis. arXiv preprint arXiv:2505.11334, 2025. 8

---

> > ### Comment · Reviewer_fPmK · 2025-08-04
> >
> > Thanks to the authors for their rebuttal and new experiments. I also read other reviews which partially share similar concerns with me. I'm not sure if the new results provided in (1) and (2) are not conclusive enough to highlight a clear contribution. I'd like to discuss it with other reviewers to make sure that I'm not missing anything.
> >
> > I don't have any other questions or requests. I'll finalize my decision and revise my score after the discussion period.

---

> > > ### Author Response · Authors · 2025-08-04
> > >
> > > Dear Reviewer fPmK:
> > >
> > > We truly appreciate your time and effort in reviewing our paper.
> > >
> > > If there are any further points that require clarification or revision, we are fully committed to addressing them promptly.
> > >
> > > Thank you once again for your patience and invaluable feedback.
> > >
> > > Best regards,

---

### Official Review · Reviewer_n6UW · 2025-07-03

**Clarity:** 2
**Significance:** 3
**Originality:** 2
**Rating:** 4
**Confidence:** 3

**Summary:**

This paper proposes WaveAR, which is a novel autoregressive (AR) framework for stochastic human motion prediction (SHMP). It applies the continuous autoregressive generation paradigm to the field of human motion prediction. It aims to solve the problem of detail loss and training instability caused by Vector Quantization (VQ) in the existing AR model when dealing with continuous input, and captures fine motion details by introducing wavelet transform.

**Questions:**

-  In the ablation study "w/o DWT" results indicate the effectiveness of the DWT module. Could the authors provide a more in-depth analysis, perhaps through visualizations or other quantitative metrics, to demonstrate how DWT specifically captures and leverages high-frequency details to improve motion prediction accuracy.

-  The paper emphasizes WaveAR's computational efficiency, but specific inference time data (e.g., milliseconds per frame) or model size (number of parameters) compared to SOTA baselines are not provided.

**Ethical Concerns:**

["NO or VERY MINOR ethics concerns only"]

**Final Justification:**

The authors have successfully addressed my concerns. After reviewing the other reviewers' comments, I believe this work features comprehensive experiments. However, the claim of novelty could be clearer. While there is an uncompared work, its unpublished status means it should not be a reason for rejection. I will maintain my "Borderline Accept" recommendation.
I will keep ``Borderline  accept``.

**Limitations:**

The authors acknowledge some limitations in their supplementary material , specifically mentioning the trade-off between accuracy and diversity, which results in relatively lower diversity scores. Building on these points.
Additional limitations can be seen in Questions and Weakness sections.

**Paper Formatting Concerns:**

No concerns about paper format.

**Quality:**

3

**Strengths And Weaknesses:**

**Strength**
- The experimental results are truly impressive. WaveAR consistently achieves the most accurate predictions (lowest ADE and FDE) on standard HMP benchmarks like Human3.6M and HumanEva-I, significantly outperforming existing state-of-the-art methods.
- WaveAR's flexible masking strategy extends its utility beyond just motion prediction; it excels in motion in-betweening tasks as well, showcasing its strong generalization capabilities across different application scenarios.

**weakness**

- In the introduction section, the author claims that WavAR is one of the "first successful application" as a contribution point, but the first one should not be an academic contribution, Instead, it should highlight the advantages of continuous autoregressive generation paradigm in this task. I think the author should re-summarize this innovation point.

- The paper mentions that WaveAR maintains "fast inference speed" and "computational efficiency". However, beyond comparative results with existing methods in Table 1, specific inference time data are not explicitly provided. A detailed discussion on the model's performance and potential bottlenecks in practical real-time applications such as robotics control or VR would greatly enhance the paper's completeness.

---

> ### Author Rebuttal · Authors · 2025-07-31
>
> To Reviewer n6UW
> ===
> We sincerely appreciate the valuable and insightful feedback from Reviewer n6UW. We will address all the concerns and polish our paper accordingly.
>
> **Weakness (1). advantages of continuous autoregressive generation paradigm**
>
> First, continuous generation avoids the inherent limitations of vector quantization (VQ) (e.g. the loss in mapping continuous motion sequences to discrete tokens[1], and the difficulty in optimizing the training process due to notorious codebook collapse[2]). Second, autoregressive generation has the following advantages: 1) The motion sequences are temporal, which is a natural fit with autoregressive modeling. 2) Better ability to capture long-distance dependencies and better ability to model long sequences. 3) Faster training and inference speed for autoregressive generation compared to diffusion's modeling of global distributions.
> Thanks for your valuable suggestion about re-summarizing the innovation point. We will change this part in the revised version.
>
> **Weakness (2). inference speed computational efficiency**
>
> Thank you for your suggestion. We evaluated inference time and parameter count on an NVIDIA V100 GPU by comparing three model variants of different sizes. Thanks to the reduced number of denoising steps, our model achieves faster inference while maintaining high prediction accuracy.
> | |Params|Inference time(s)|APD↑|ADE↓|FDE↓|
> |:----:|:----:|:----:|:----:|:----:|:----:|
> |HumanMAC|28.4M|1.25|6.301|0.369|0.480|
> |Ours(tiny)|26.8M|**0.21**|**4.952**|0.354|0.465|
> |Ours(small)|51.9M|0.43|4.458|0.347|0.452|
> |Ours(base)|86.5M|0.65|4.633|**0.342**|**0.450**|
> We will add it in the revised version.
>
> **Questions (1). in-depth analysis of DWT**
>
> First, we visualize the spectrum of WaveAR w/o and w/ DWT, respectively. Since it is not possible to add images in the rebuttal, we count the percentage of high-frequency component averages. We randomly sampled 200 instances, and w/ DWT, the average proportion of the amplitude of high-frequency components is 0.2429, whereas w/o DWT, the average proportion of the amplitude of high-frequency components is 0.1624. This demonstrates that the DWT effectively captures the high-frequency component of the motion.
> Second, we found some “static” phenomena in the generated motion, i.e., the person is hardly moving. After applying DWT, these cases will generate motion with more details instead of "static", , demonstrating that high-frequency information leads to richer motion details. Since it is not possible to add images in the rebuttal, we will update this part of the visualization to the supplementary materials in the revised version.
> We also conduct an experiment on the ablation between DCT and Wavelet module. The results are shown below. We can see that incorporating spectral information improves motion prediction accuracy, and that DWT—by capturing high-frequency details in the input beyond what DCT can—yields even more precise predictions.
> | |APD|ADE|FDE|MMADE|MMFDE|
> |:----|:----|:----|:----|:----|:----|
> |w/o DWT&DCT|4.781|0.381|0.503|0.535|0.549|
> |w/ DCT|**5.016**|0.362|0.478|0.521|0.537|
> |w/ DWT|4.458|**0.347**|**0.451**|**0.513**|**0.535**|
>
> **Questions (2). inference time and model size**
>
> Please refer to Weakness(2).
>
>
> **References:**
>
> [1] Chuan Guo, Yuxuan Mu, Muhammad Gohar Javed, Sen Wang, and Li Cheng. Momask: Generative masked modeling of 3d human motions. In Proceedings of the IEEE/CVF Conference on Computer Vision and Pattern Recognition, pages 1900–1910, 2024. 1, 2, 4
>
> [2] huanxia Zheng and Andrea Vedaldi. Online clustered codebook. In Proceedings of the IEEE/CVF International Conference on Computer Vision, pages 22798–22807, 2023. 1

---

> > ### Comment · Reviewer_n6UW · 2025-08-04
> >
> > Thanks for the detailed response. I don't have any additional questions. I will finalize my decision and keep my original rating score.

---

> > > ### Author Response · Authors · 2025-08-06
> > >
> > > Dear Reviewer n6UW:
> > >
> > > Thank you for your positive and encouraging feedback on our work! We're delighted to hear that our rebuttal has addressed your concerns. We sincerely appreciate the time and effort you’ve invested in providing detailed reviews and valuable suggestions to help improve our work.
> > >
> > > Best regards,

---

### Note · Authors · 2025-08-14

We sincerely thank all reviewers and the AC for their time, effort, and thoughtful feedback on our paper. Your constructive comments have been invaluable in improving our work.

Our paper presents WaveAR, a continuous autoregressive framework for stochastic human motion prediction. By combining a  Spatio-Temporal VAE with multi-scale wavelet-guided masked autoregressive diffusion, WaveAR eliminates quantization artifacts, preserves fine motion details, and achieves state-of-the-art accuracy with fast inference on widely used datasets.

In our rebuttal, we addressed the key concerns as follows:

●AMASS large-scale dataset experiments – We added experiments on the AMASS dataset, showing that our model still achieves the most accurate prediction results.

●Inference speed and computational efficiency – We conducted additional experiments with different model sizes to analyze the trade-offs between accuracy and efficiency.

●DWT module analysis – We provided more detailed ablation studies and evaluations for the newly proposed DWT module.

●Limitations and clarity of presentation – We analyzed the trade-off between diversity and accuracy, and will revise the manuscript in the camera-ready version to improve clarity and language precision.

In summary, we have carefully addressed **all major concerns** raised by the reviewers, strengthened the experimental evidence and discussed limitations more explicitly. We once again thank all reviewers and the AC for their valuable input, which has significantly improved the quality of our work. We hope that our responses and the additional results will satisfy the concerns raised and earn your acceptance.

---

### Decision · Program_Chairs · 2025-09-17

**Decision:**

Accept (poster)

**Comment:**

This paper introduces WaveAR, an autoregressive framework for stochastic human motion prediction (SHMP) that operates in a continuous latent space without relying on vector quantization. WaveAR consists of two main stages: a lightweight Spatio-Temporal VAE (ST-VAE) compresses raw 3D-joint sequences into continuous latent tokens, and a masked autoregressive diffusion model, guided by multi-scale wavelet features, predicts future poses. The wavelet features, extracted via a 2D discrete wavelet transform, capture both low- and high-frequency motion details, enhancing prediction accuracy. A fusion module combines these spectral cues with temporal context using alternating cross-attention and self-attention layers. Experiments on Human3.6M and HumanEva-I benchmarks demonstrate that WaveAR achieves state-of-the-art performance in accuracy and efficiency.

The integration of continuous autoregressive modeling with wavelet transforms is a clear advancement over prior discrete or DCT-based approaches. Ablations validate the contributions of DWT and ST-VAE, showing improved high-frequency detail capture. WaveAR shows superior accuracy metrics and faster inference compared to baselines. However, the reviewers have raised several concerns. The model prioritizes accuracy over diversity, which may limit its use in some scenarios. Initial claims about being "first" in continuous autoregressive modeling were overstated, as similar techniques exist in motion synthesis. Also, it lacks a quantitative comparison against MotionWavelet.

During the rebuttal phase, the authors have made great efforts in addressing the concerns. For the concern of missing large-scale dataset validation raised by Reviewers p7mw and fPmK, the authors added AMASS results showing superior accuracy. For the concern of DWT vs. DCT ablations raised by Reviewer 76BV, the authors showed DWT’s superiority in capturing high-frequency details. For the concern of inference speed raised by Reviewer n6UW, the author provided inference times and model sizes. Most reviewers are satisfied with the rebuttal. After the discussion, all the reviewers agree that the comparison with MotionWavelet should not be grounds for rejection, as it is an unpublished work.

The paper’s technical merits, coupled with thorough rebuttal responses, outweigh its limitations. The authors are encouraged to improve the paper by considering the reviewers’ comments carefully.